# Study of Discharge Characteristics on Ignition Performance via High-Speed Imaging in a CVCC

Qingchu Chen [1] , Tatsuya Kuboyama [1,*], Yasuo Moriyoshi [1] and Kazuhiro Oryoji [2]

1 Graduate School of Engineering, Chiba University, 1-33, Yayoicho, Inage-ku, Chiba-shi 263-8522, Chiba, Japan; chenqcseven@chiba-u.jp (Q.C.); ymoriyos@faculty.chiba-u.jp (Y.M.)
2 Hitachi Ltd., Research and Development Group, 7-1-1, Omika-cho, Hitachi-shi 319-1292, Ibaraki, Japan; kazuhiro.oryoji.ax@hitachi.com
* Correspondence: tkuboyam@faculty.chiba-u.jp; Tel.: +81-43-290-3916

**Abstract:** Advanced combustion technologies, like highly boosted and lean or dilute combustion, have been employed to meet the demands of high efficiency and low emissions in SI engines, which have increased the challenges of ignition control. It is essential to find a suitable ignition strategy due to the need to develop a next-generation spark ignition system. In this study, simultaneous visualization by a high-speed infrared camera (FLIR X6900sc) and a conventional high-speed camera (FASTCAM SA-X) is carried out to obtain deeper insights into the ignition process in a constant volume combustion chamber (CVCC). Infrared images have provided a more accurate way of measuring the initial flame and are able to analyze quantitatively. Ignition performance is studied with various mixture dilutions, flow conditions, and discharge characteristics. Two types of ignition coils that have the same discharge energy were analyzed in particular. The results show that extending the discharge duration is more helpful in improving the ignition performance under the increasing dilution ratio, compared to the enhanced discharge current at the same discharge energy. However, the discharge current plays a more vital role in perfecting the ignition performance under the increasing local flow velocity than the discharge duration.

**Keywords:** discharge characteristics; ignition performance; simultaneous visualization

## 1. Introduction

The burden of traditional resources and environmental concerns demand high efficiency and low emissions in internal combustion engines (ICEs) for on-road vehicle applications. As spark-ignition (SI) engines fueled with gasoline continue to be the primary power system for light-duty vehicles, there is a need to increase the thermal efficiency and decrease the exhaust emissions for the SI engine. In recent years, several advanced combustion technologies like highly boosted and lean or dilute combustion had increased the challenges of ignition control in SI engines [1,2]. Prechamber igniter, as a new method, is also challenging the traditional SI systems, and Corrigan et al. make a comparison between the prechamber and the standard SI systems, revealing that the knock-limited combustion phasing has improved in prechamber igniter method [3]. Therefore, the development of a next-generation spark ignition system is greatly needed to meet these challenges.

A suitable ignition strategy will play a central role in optimizing the ignition system. Several studies were conducted to investigate the ignition performance in SI engines. Naoto et al. had employed a variable control device of discharge current instead of the conventional ignition system, and they suggested that, ideally, the ignition system should control the discharge current according to engine conditions [4]. Alger et al. used a high-energy continuous discharge ignition system to show that a higher discharge current can help resist blowouts and improve EGR tolerance [5]. The effects of ignition on combustion variation in SI engines had been studied. Combustion variation is affected by the local spatial distribution of mixture concentration around the spark plug [6,7] and the spark

ignition position and timing [8]. Furthermore, it is well known that such combustion instability significantly impacts the subsequent flame propagation and pollutant emissions.

To provide a deep insight into the discharge characteristics of the spark plug in the cylinder, the optical diagnostics method is considered an effective way to understand the ignition process, and lot of research has been conducted by using either optically accessible engines or the endoscope. Atsushi et al. used a conventional ignition system to investigate the flow velocity and discharge channel behavior in an optical SI engine. They observed that higher flow velocities increase the discharge channel stretching the length and decrease in the ignition delay [9]. Chang et al. investigated the discharge channel shortening phenomena including the short-cut and restrike under both ambient air and stoichiometric conditions, and results have shown that the new short circuit of the discharge channel is more easily formed inside the original discharge channel circuit under stoichiometric conditions [10]. Oryoji et al. investigated the ignition phenomena using an endoscope to realize the in-cylinder optical measurement. It demonstrated that the initial combustion period correlated with spark stretch before the 1st restrike and spark-stretch rate [11,12]. Tawfik et al. used the optical engine to study the effects of the spark plug gap on the flame kernel growth and engine performance by the method of PLIF, and as a result, the flame kernel growth area increased with the spark plug gap increases, while the engine performance increased slightly due to the reduction in cyclic fluctuation as the spark plug gap increased [13]. Peterson et al. also used the method of high-speed PLIF and PIV to analyze the spatial and temporal evolution of the fuel distribution on flame kernel development. Results have helped a better understanding of the nature of poor burning cycles at each dilution level [14–16].

In addition, due to the limit of the engine bench test, the initial flame formation during the ignition process still had not been studied very well. Moreover, the in-cylinder ambient environment parameters, such as gas flow velocity, gas composition, and ambient pressure et al., are challenging to separate and analyze independently due to the complexity of such parameters interactions. On the other hand, the mixture concentration in the vicinity of the spark plug is not always the same at every firing cycle, and under the actual engine bench test, the ignition swing will also alter the indicating parameters of engine performance. While the experiments in a constant volume combustion chamber (CVCC) help fill in the gap in this aspect. Therefore, to understand the effects of individual factors on the ignition process, it is necessary to carry out the experiments in a constant volume combustion chamber (CVCC) to provide a deep insight into the discharge characteristics at various ambient conditions.

Zhenyi Yang et al. used the schlieren photography method to investigate the initial flame formation. The results indicated that an attached flame kernel could be formed by either increasing the discharge current or prolonging the discharge duration under the flow velocity of 25 m/s [17]. Zhang et al. demonstrated that multi-strike discharge was able to generate kernels that merge and lead to initial flame propagation [18]. Suzuki et al. discovered that a high current was necessary to maintain the discharge channel extension under the strong flow condition [19]. However, the lower flow conditions were still not clear. Anton et al. had demonstrated that increasing the discharge duration resulted in the increase in the maximum length of the discharge channel by a specially designed arc test rig [20]. However, the actual discharge current duration was about 0.3 ms to 0.9 ms, shorter than the conventional ignition system.

Furthermore, the technologies of cooling infrared cameras have been improved in recent years and are accessible in many fields. Mancaruso et al. applied a high-speed infrared camera to the bottom view of an optical visualization diesel engine to observe the spray combustion of biofuels in the CO2 and HCs bands. Due to this, it is possible to detect the reaction immediately after SOC of the pilot injection and after the end of visible combustion in the infrared range. Furthermore, it is possible to evaluate the spray evaporation and mixing process, and the combustion evolution after the main injection [21]. Besides, Okabe et al. measured the temperature of spark plugs as an example of applying

the near-infrared region to a spark-ignition engine. The results indicated that the factors of pre-ignition depend on the engine operating conditions [22].

In general, the discharge characteristics, such as discharge current and duration, have been widely studied in the literature, however, the ignition energy was changed when only focusing on the discharge current or the discharge duration in the previous studies. Under the same ignition energy, the roles of discharge current and duration of the ignition performance at various ambient conditions is still unclear. Therefore, this study uses three types of ignition coils to conduct the test, two of which have the same ignition energy. Moreover, combustion products such as CO and $CO_2$ have a specific absorption band in the infrared region. There is no report of detecting the initial flame within the CO infrared region during the ignition to the author's knowledge. Hence, this study used the simultaneous visualization method of high-speed infrared and natural imaging in the constant volume combustion chamber to provide the initial flame formation and discharge channel behaviors during the ignition process.

## 2. Experimental Setup

### 2.1. Constant Volume Combustion Chamber

The experimental study has been carried out in a cubical optically accessible constant volume combustion chamber (CVCC), as shown in Figure 1. The CVCC features a pancake internal combustion chamber with a 60 cm$^3$ internal volume.

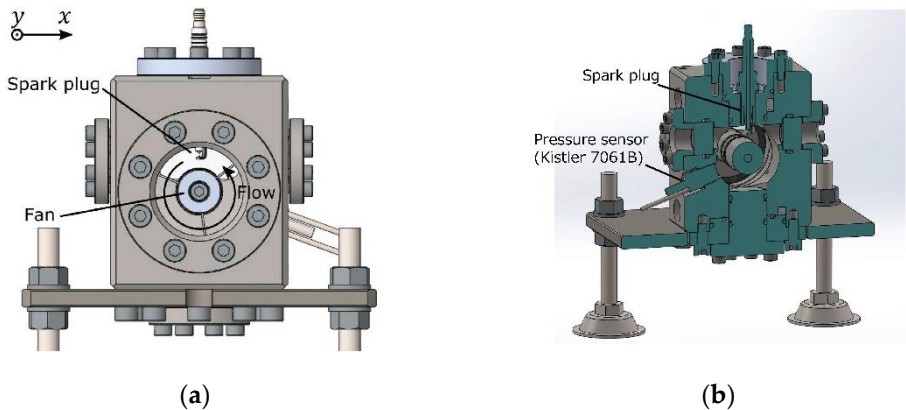

(**a**)　　　　　　　　　　　　　　　　　　　　(**b**)

**Figure 1.** Constant volume combustion chamber. (**a**) Front view; (**b**) Cross-sectional view.

Due to the small volume, the pressure variation can be detected by a highly sensitive pressure sensor, which was installed at the lower left section of the chamber. Four pieces of sapphire windows are mounted in four directions of the cubical CVCC. A fan for generating cross-flow is configured at the rear side of the chamber. The flow intensity can be varied by controlling the rotating speed of the fan. A commercially available spark plug (NGK DILKAR7C9H) with a 0.9 mm electrode gap is installed at the top of the CVCC. Besides, the stainless-steel body of the CVCC is equipped with a coolant channel that can maintain the temperature of the CVCC by controlling the coolant temperature.

### 2.2. Simultaneous Visualization Method

Most researchers use high-speed photography to investigate the discharge channel behaviors and the initial flame kernel during the ignition process. The discharge channel is relatively easy to detect directly by natural imaging since the discharge channel typically emits a visible glow. However, the initial flame kernel is not easy to detect as it is invisible on a visible band [23,24]. The Schlieren photography method has been widely used to investigate the initial flame kernel in the combustion chamber. However, this method is hard to distinguish between the preheated zone and initial flame, as the Schlieren method visualizes the density gradient in the optical path as a shadow. The preheat zone caused by the discharge channel can be also visualized. Even under no fuel conditions, the preheated zone was still detected by the Schlieren imaging [25], which has affected the accuracy of the

initial flame kernel area. Therefore, in this study, a high-speed infrared camera was used to detect the initial flame as most of the combustion products, such as carbon monoxide (CO), carbon dioxide ($CO_2$), and water ($H_2O$), absorb specific infrared wavelengths and radiate the infrared when the components turn to be hot. As shown in Figure 2, the infrared camera is a sensor-cooled infrared camera equipped with an indium antimonide (InSb) element, and it can store 1/4-inch filters inside. It is possible to shoot at 1000 fps with total pixels (640 × 512 px) and at a maximum of about 30,000 fps.

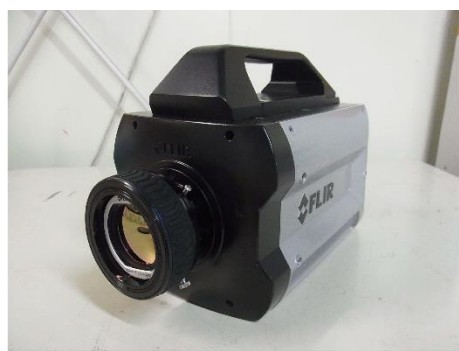

**Figure 2.** Infrared high-speed camera (FLIR X6900sc).

Simultaneous visualization method was carried out by a high-speed infrared camera (FLIR X6900sc) and a conventional high-speed camera (Photron SA-X) in this study, as shown in Figure 3. The infrared dichroic mirror is utilized to reflect the light with 3000~ nm wavelength and pass the light with 400~700 nm wavelength. According to the NIST Chemistry WebBook [26], as shown in Figure 4, CO has two high absorbances around 4600 nm and 4700 nm in the spectral range compared with $CO_2$ and $H_2O$. Therefore, the bandpass 4560 ± 135 nm filter is utilized in the high-speed infrared camera for capturing the infrared radiation images of carbon monoxide with 5000 frames per second (fps). The conventional high-speed camera is set up for taking the natural imaging of the discharge channel with 20,000 fps in the visible band.

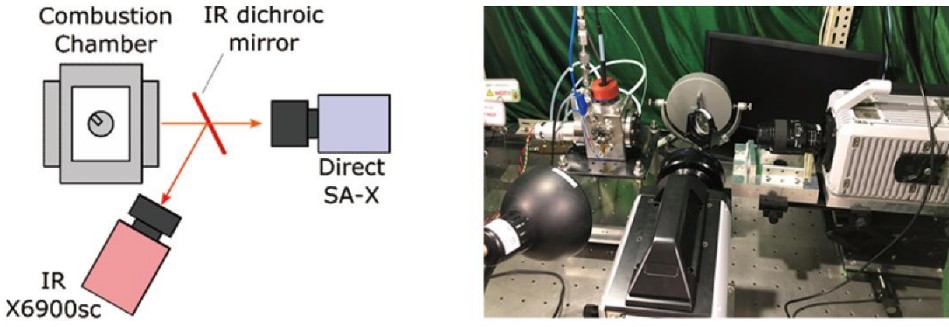

**Figure 3.** Schematics of the imaging method.

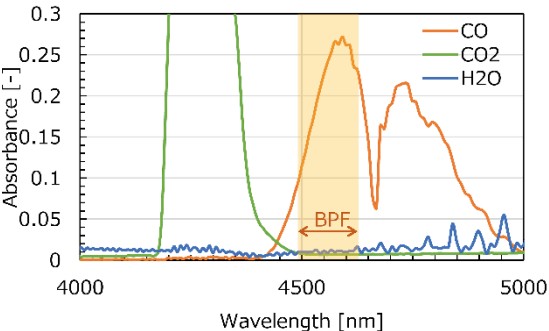

**Figure 4.** IR absorbance spectrum.

In addition, to confirm the preheat zone of infrared images, the test was conducted under both fuel and no fuel conditions. Under no fuel condition, there was no initial flame kernel generated. Still, the preheat zone will change the temperature at the vicinity of the spark plug via the discharge channel formation. Hence, it is necessary to confirm whether the preheat zone affects the filtered signal in the infrared images or not. As shown in Figure 5, the upper part is the no fuel condition, the lower part is the fuel condition. In each part, the top line is the CO radiation images taken by the infrared camera, and the lower line is the direct images taken by the conventional high-speed camera. The brightness of the initial flame kernel is saturated in the infrared image under fuel condition, while under the no fuel condition, the brightness of CO radiation is very weak and can hardly be confirmed between the spark plug gaps, and it can be found that the preheat zone by the discharge channel can be ignored compared with the initial flame kernel. Thus, the CO radiation images can detect the area of the initial flame kernel more accurately than the Schlieren imaging.

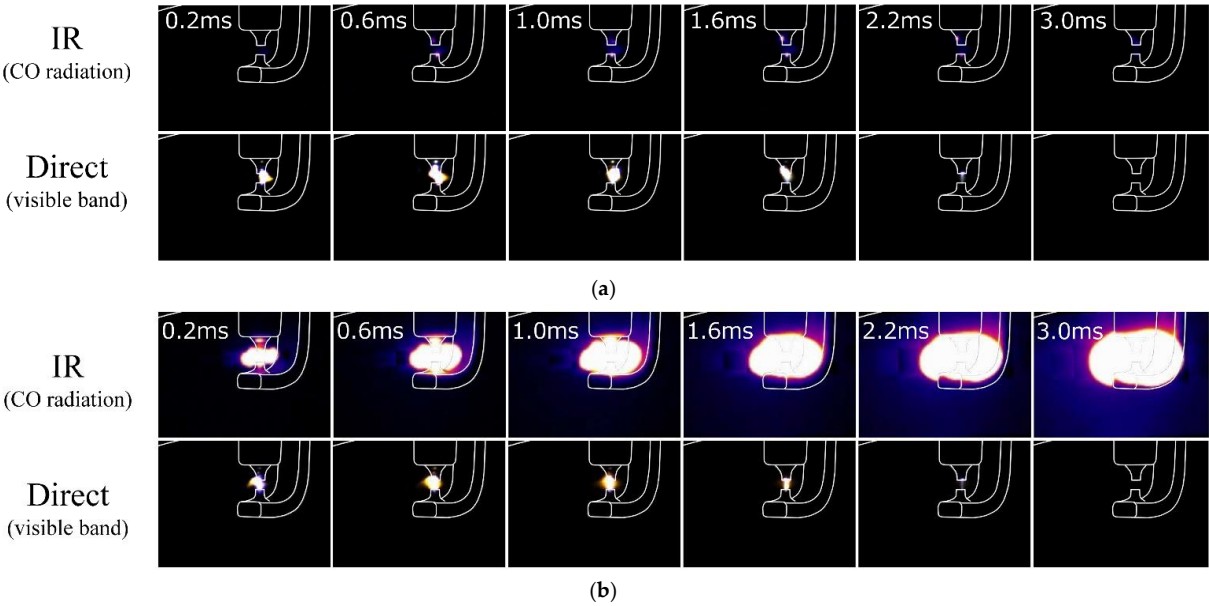

**Figure 5.** Direct and CO radiation images on static fields with/without fuel conditions (top: CO radiation images (aim to detect the initial flame kernel)/lower: direct images (aim to capture the discharge channel). (**a**) With no fuel condition; (**b**) With fuel condition.

### 2.3. Ignition Coils and Test Conditions

Figure 6 shows the discharge characteristics of three types of ignition coils. This research takes coil A as the standard coil, coil B as the high current coil, and coil C as the high energy coil. The detailed specifications of the coils are shown in Table 1. As described in Table 1, coil A and coil B have almost the same ignition energy, and coil C is the highest energy one. However, coil C has almost the same discharge duration as coil A, and coil B is much shorter than coil A and coil C. Coil B and coil C had almost the same discharge current, which is much higher than coil A.

The experiments were carried out on three conditions: basic condition, high dilute condition, and high flow condition to determine the suitable discharge strategy under different conditions. Experimental conditions are as listed in Table 2. In order to control the composition of the mixture, we used the $O_2$ and $N_2$ to imitate the air and added more $N_2$ to imitate the dilute condition, the air-propane mixture with an equivalence ratio of one is used in the experiment. The combustion chamber temperature was fixed at 358 K and initial pressure of four bar for ensuring the same initial conditions of each test. Steady flow velocities were calculated by 3-D numerical simulations of CONVERGE software. Each condition of each coil was tested ten times to eliminate the experimental error.

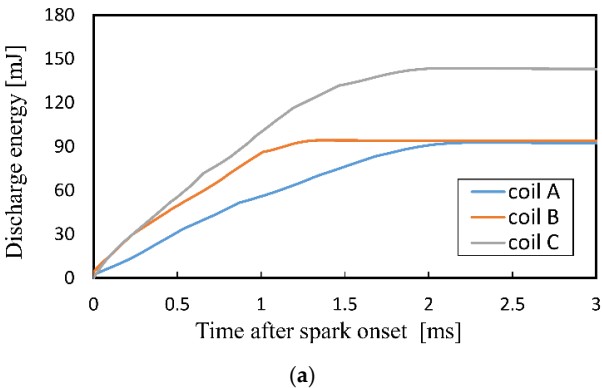
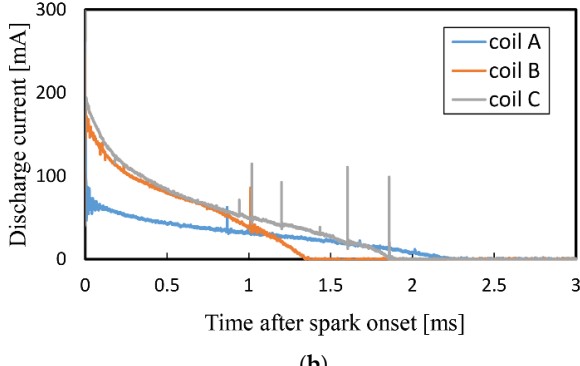

(**a**)                                                    (**b**)

**Figure 6.** Discharge characteristics of the ignition coils. (**a**) Discharge energy; (**b**) Discharge current.

**Table 1.** Detail specifications of the ignition coil.

|                           | Coil A | Coil B | Coil C |
| ------------------------- | ------ | ------ | ------ |
| Discharge current [mA]    | 65     | 170    | 180    |
| Discharge duration [ms]   | 2.2    | 1.3    | 1.9    |
| Ignition energy [mJ]      | 95.0   | 95.0   | 165.0  |

**Table 2.** Summary of experimental conditions.

| Conditions | Basic Condition | High Diluted Condition | High Flow Condition |
| --- | --- | --- | --- |
| Nitrogen ($N_2$) dilution | 24% | 28% | 24% |
| Mixture composition | $C_3H_8$: 3.06%, $O_2$: 15.29%, $N_2$: 81.65% | $C_3H_8$: 2.90%, $O_2$: 14.49%, $N_2$: 82.61% | $C_3H_8$: 3.06%, $O_2$: 15.29%, $N_2$: 81.65% |
| Flow velocity | 8 m/s | 8 m/s | 18 m/s |
| Equivalence ratio | | 1 | |
| Initial temperature | | 358 K | |
| Initial pressure | | 4 Bar | |

## 3. Results and Discussion

### 3.1. Combustion Results

Figure 7 shows the ignition probability of all the coils under different conditions. Ignition probability is the ratio of the successful ignition cases and the total number of experimental cases (10 times). The blue column stands for coil A (standard coil), the orange column denotes coil B (high current coil), and the gray column represents coil C (high energy coil). According to Figure 7, it can be observed that three types of coils have the same ignition probability ratio at the basic condition. There is no misfire case under the basic condition for all the coils. However, the differences among the three coils occur when increasing the mixture dilution ratio or enhancing the flow velocity, high current coil B gets the lowest ignition probability when under the high diluted condition compared to other coils, while standard coil A and high energy coil C still have no misfires cases under such conditions. When under the high flow condition, standard coil A gets the lowest ignition probability, while high energy coil C gets the highest ignition probability. High current coil B is better than the standard coil A. It reveals that the high energy coil C resulted in high level ignition probability under all the conditions. This means that enhancing the ignition energy can improve the ignition probability remarkably. However, high energy will increase the wear and tear of the spark plug due to the increased discharge current and prolonged discharge duration, hence, coil A and coil B with the same lower ignition energy need to be considered in particular. Standard coil A with a longer discharge duration has a better ignition probability under the high diluted condition. In contrast, high current coil B with a higher discharge current has a better ignition probability under the high flow

condition. Ignition probability can reflect the ignition performance by and large, when comparing results of the basic condition and the high diluted condition, due to the lowest ignition probability ratio of coil B under the high diluted condition, and, one thing that can be found is that the discharge duration is more important than the discharge current with regard to affecting the ignition probability when only increasing the mixture dilution ratio; when focusing on the results of the basic condition and high flow condition, it is notable that the discharge current becomes the dominant factor compared to the discharge duration when enhancing the gas flow velocity, since coil A gets the lowest ignition probability.

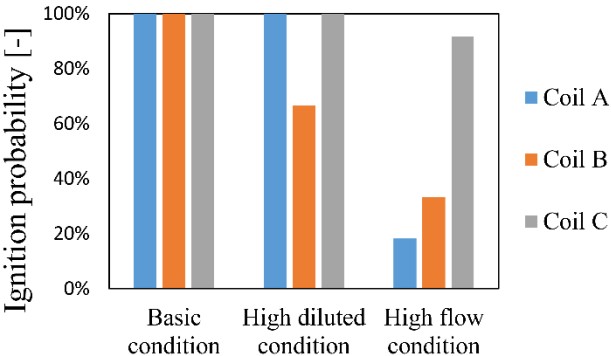

**Figure 7.** Ignition probability of all the conditions.

Due to the small internal volume of the combustion chamber, the small pressure changes can be detected by a highly sensitive pressure sensor, and the results of ignition delay are shown in Figure 8. Ignition delay is defined as the time period between the start of discharge to the MFB1 (mass fraction burn of 1 percent). It can be found that coil B with a high discharge current results in the longest ignition delay in the high diluted condition, while when in the high flow condition, coil A with the extended discharge duration results in the longest ignition delay. The difference in initial flame formation is largely affected by ignition. Hence, given that coil A has the same ignition energy as coil B, it shows that coil A with extended discharge duration favors more in improving the ignition performance than coil B with higher discharge current under high diluted conditions. Nevertheless, the opposite of this result is the case under high flow velocity conditions, as the higher discharge current is more important under the high flow condition.

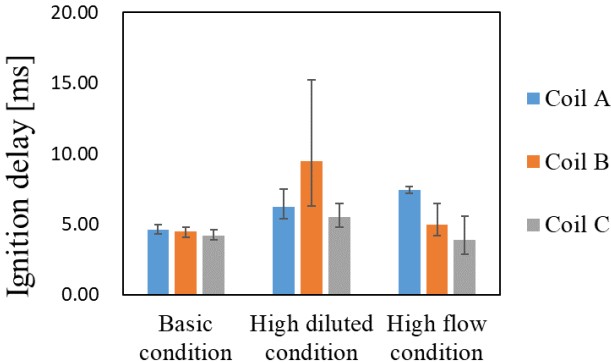

**Figure 8.** Ignition delay of all the conditions.

According to the R–W (Rasswei–Withrow) model [27], the pressure changes are only caused by the combustion in the constant volume combustion chamber since the volume is constant in it, the heat release rate can consequently be calculated from the pressure data. Figure 9 illustrates the heat release at different conditions of all the experimental cases and about three types of coils. As shown in Figure 9, combustion time is defined as the time period from the start of discharge to the point of peak heat release. The graph of heat release can reflect the combustion speed via the length of combustion time and

embodies the combustion stability by the fluctuation of combustion time of all the cases. From Figure 9a of the basic condition, all the coils have no significant difference from each other in heat release histories, additionally, the combustion speed and combustion stability have a high level of all the cases; From Figure 9b of the high diluted condition, combustion becomes slower and unstable as compared with the basic condition, the timing of the peak heat release point is put off, which means the combustion time is extended. Furthermore, the combustion fluctuation case-to-case is increased, and the high current coil B has the slowest combustion case and even has misfire cases under the high diluted condition. Although standard coil A and high energy coil C have the same ignition probability, coil C combustion cases keep a smaller combustion fluctuation than that of coil A. In Figure 9c of the high flow condition, the combustion time of the fastest case of coil C shows a slightly shorter compared with the fastest case of coil C under the basic condition, although the combustion fluctuation is increased, which reveals that increasing the flow velocity may improve the combustion speed only when the ignition energy is increased correspondingly.

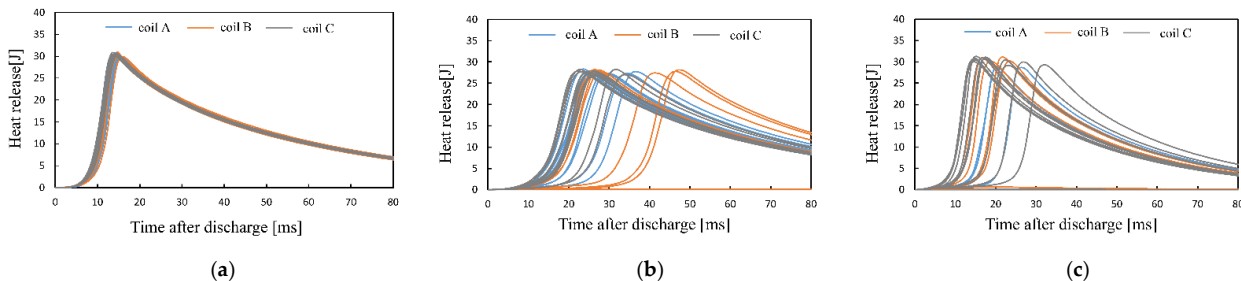

(a)  (b)  (c)

**Figure 9.** Heat release under different conditions: (**a**) Basic condition; (**b**) High diluted condition; (**c**) High flow condition.

As described in the combustion results, several interesting results can be found. On the one hand, increasing the mixture dilution ratio will decrease the combustion speed. Even increasing the ignition energy cannot reach the combustion speed of low diluted condition, when focusing on the same ignition energy coils, prolonging the discharge duration is more important than increasing the discharge current for the ignition performance. On the other hand, increasing the flow velocity may improve the combustion speed only if increasing the ignition energy correspondingly. Additionally, if the ignition energy is constant, increasing the discharge current has more benefits than prolonging the discharge current for improving the ignition performance.

Since the experiments were carried out in the combustion chamber, the combustion results were mainly affected by the ignition process. Therefore, in order to explain the reason for combustion results, a detailed investigation of simultaneous visualization for the spark discharge channel behaviors and initial flame kernel formation was carried out during the ignition process. The results are shown in the next section.

### 3.2. Simultaneous Visualization Results

Figure 10a–c shows the simultaneous visualization results under different experimental conditions. Each figure contains three rows of images representing three types of coils, and each row of images includes two kinds of photographs. The top one shows the CO radiation images taken by the high-speed infrared camera and the lower one shows the direct photographs taken by the normal high-speed camera. The profile of the spark plug was drawn with a white line to show the location of the spark plug.

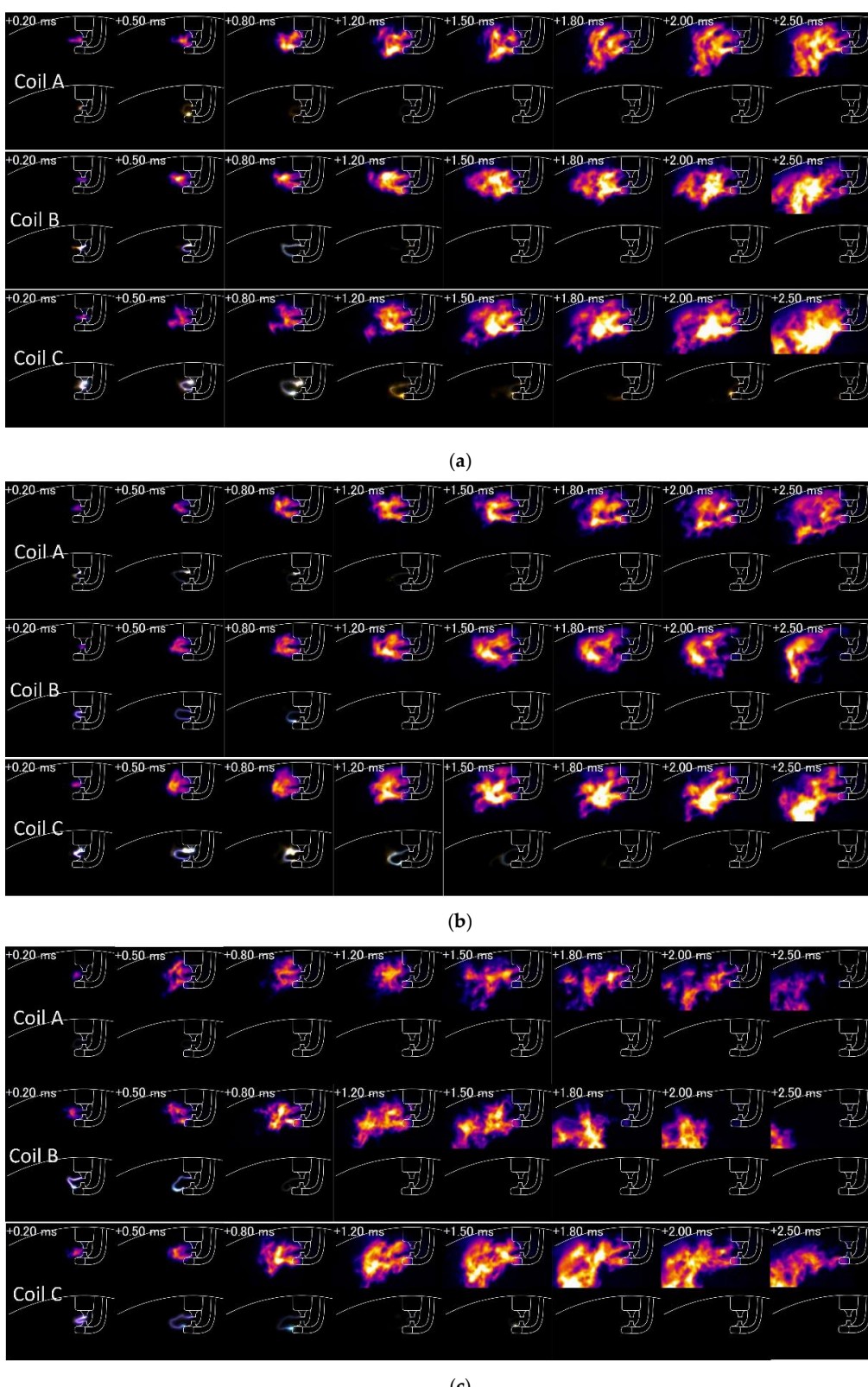

**Figure 10.** Simultaneous visualization results of different ignition coils (upper: CO radiation images/lower: direct photographs. Basic condition with the $N_2$ dilution of 24% and flow velocity of 8 m/s, high diluted condition with the $N_2$ dilution of 28% and flow velocity of 8 m/s, high flow condition with the $N_2$ dilution of 24% and flow velocity of 18 m/s): (**a**) Basic condition; (**b**) High diluted condition; (**c**) High flow condition.

According to Figure 10, it can be found that after the electrical breakdown, a discharge channel was formed at the spark plug gap, meanwhile, the initial flame was generated immediately along with the discharge channel. Then, the discharge channel was stretched downstream with the flow, and the initial flame was developed during the ignition process.

As shown in Figure 10a, high current coil B and high energy coil C have the higher luminance of the discharge channel during the ignition process. Furthermore, after 1.5 ms, the initial flame propagates farther than the standard coil A. However, all the coils are holding the initial flame around the spark plug at 2.5 ms. It is found that all the coils can form an attached flame kernel after the discharge duration under the basic condition. This means all the coils in this study can produce the self-sustained flame after the discharge process, so that the flame grows and propagates rapidly. Therefore, the combustion can complete with a shorter duration and high stability.

Under the high diluted condition, according to Figure 10b, the initial flame of high current coil B is detached from the spark plug at 2.5 ms, while the standard coil A and high energy coil C can hold the flame attached. Comparing standard coil A with high current coil B, the initial flame of coil B propagates farther than coil A, at 1.5 ms, due to the higher current. However, the propagation distances of the initial flame front are almost the same after 2.0 ms, and the rear of the flame moved downstream. Hence, it is more difficult to form the self-sustained flame for coil B under the low flow and high diluted conditions, which results in a lower ignition probability and lower combustion stability. On the other hand, from the CO radiation images, the initial flame of high diluted condition develops slower than the basic condition. Even in the high energy coil C, comparing the images of basic condition and high diluted condition at 2 ms, the initial flame of the high diluted condition is smaller than that of the basic condition. This may mean that increasing the mixture dilution ratio will decrease the combustion speed, and even an increase in the ignition energy cannot reach the combustion speed of the low diluted condition. An explanation for this is that increasing the dilution ratio only influences the decrease in speed of the flame kernel formation and flame propagation, shortening the discharge duration results in the initial flame kernel being detached from the spark plug, which contributed to lowering the combustion stability. Due to the little effect on the discharge channel behaviors, the initial flame kernel was smaller than that in the low dilution ratio condition, even using the high ignition energy coil C.

When under the high flow condition, comparing the direct photographs at 0.5 ms of coil C in Figure 10a,c, it should be noticed that the discharge channel length of the high flow condition is longer than that of the basic condition. That is to say, the discharge channel stretches faster when enhancing the flow velocity. Due to the initial flame being formed along with the discharge channel, a longer discharge channel length results in the initial flame propagating farther. As shown in Figure 10c, all the initial flames detach and move downstream at 2.5 ms for all the coils, which prevented the initial flame from forming the self-sustained flame. Nevertheless, high current coil B and high energy coil C have a larger flame area at 1.5 ms than that of the standard coil A, which indicates that high current could ensure the initial flame generates under such an enhanced flow condition. While standard coil A with a lower discharge current results in a smaller initial flame kernel that disappears after the discharge process. On the other hand, the high flow velocity not only lengthens the discharge channel length, but also wrinkles the initial flame. As shown in Figure 10c of coil A, the luminance of the initial flame is lower than the basic condition (Figure 10a). Furthermore, according to Figure 10c of coil B, the initial flame starts to detach the spark plug at 1.8 ms, which is earlier than the high diluted condition (Figure 10b). However, the high energy coil C maintains the initial flame growth well for the higher luminance of the initial flame and holds the initial flame around the spark plug for more time compared to low ignition energy coils. Thus, it can be considered that increasing the flow velocity may improve the combustion speed only if increasing the ignition energy correspondingly as the flow intensity has both an influence on the discharge channel stretching and flame kernel formation.

According to the discussion above, discharge coils with different discharge characteristics affect the generation of the discharge channel and also make some contribution to the initial flame. The mixture composition and flow velocity have a great influence on the initial flame growth and propagation.

The quality of initial flame generation affects the combustion directly, and quantitative analysis of the initial flame helps support the explanation of the combustion results. Therefore, the value of the flame area and the mean luminance are measured to estimate the growth of the initial flame, and the results are shown in the next section.

### 3.3. Quantitative Analysis of the Initial Flame Area and Initial Flame Luminance

In order to get a deeper understanding of the initial flame kernel, the initial flame area and initial flame luminance were quantitatively measured for all experimental cases. The initial flame area was calculated via CO radiation image binarization. As shown in Figure 11, firstly, a threshold value was set, then the raw infrared image was binarized and the flame contour could be determined. Then the projected initial flame area was achieved by measuring the pixel of the binary image. The average value of the flame area and error bar for all the cases are shown in Figures 12a, 13a and 14a under different conditions. The mean luminance of the initial flame was calculated by the integral luminance value divided by the flame area value, and the results are shown in Figures 12b, 13b and 14b. The blue line stands for the standard coil A, the orange line represents the high current coil B, and the gray line represents the high energy coil C. Both flame area and flame luminance help to evaluate the initial flame generation and reaction intensity.

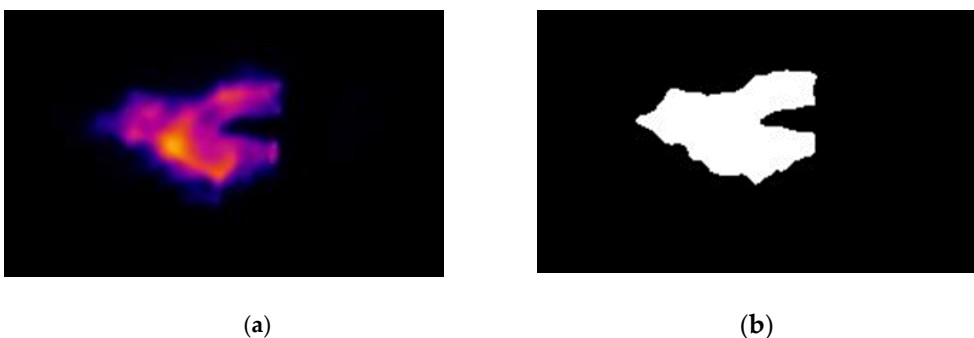

(**a**)          (**b**)

**Figure 11.** Procedure of determining the initial flame area: (**a**) Raw infrared image; (**b**) Binary image.

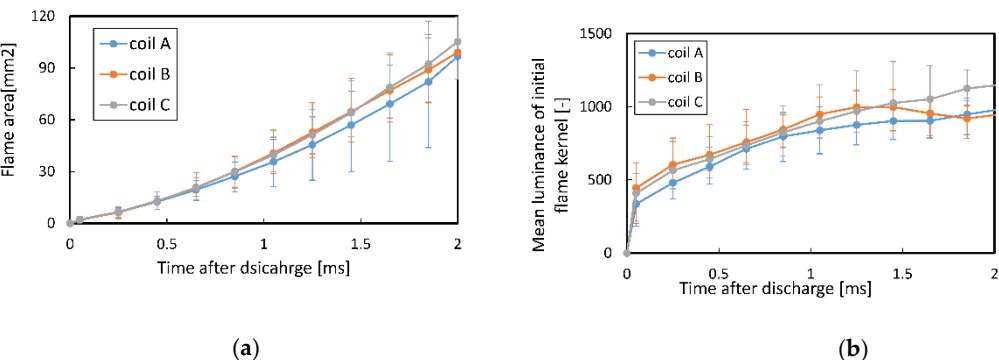

(**a**)          (**b**)

**Figure 12.** Basic condition: (**a**) Initial flame area; (**b**) Initial flame luminance.

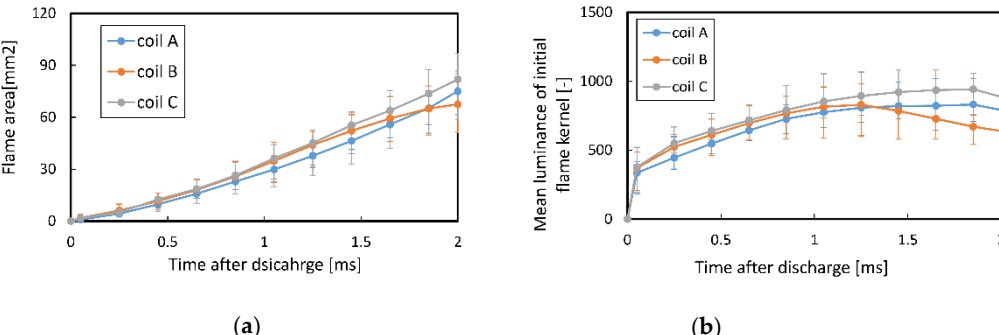

**Figure 13.** High diluted condition: (**a**) Initial flame area; (**b**) Initial flame luminance.

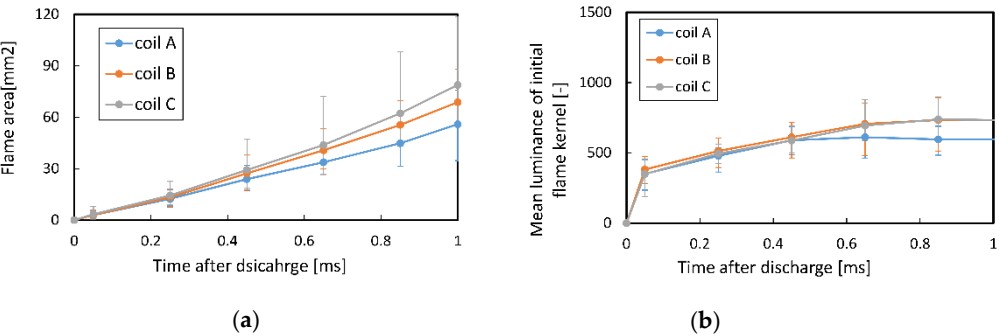

**Figure 14.** High flow condition: (**a**) Initial flame area; (**b**) Initial flame luminance.

According to Figure 12, which shows the basic condition, the high current coil B and high energy coil C generate almost the same flame area as they have the same discharge current until 1.5 ms. The speed of the initial flame growth of high current coil B decreases after 1.5 ms, while the high energy coil C keeps a high speed until 2 ms. Standard coil A has the lowest initial flame growth speed and results in a smaller initial flame size. However, it can keep the growth speed and almost gets the initial flame size at 2 ms compared with coil B. It can be explained that although a higher discharge current results in a larger initial flame, the initial flame growth slowed down after the end of the discharge of coil B (discharge duration: 1.3 ms). Coil A and coil C remain at the speed of initial flame growth as the discharge continues until 2 ms. Initial flame luminance indicates the above results more clearly. As shown in Figure 12b, the mean luminance of the initial flame for high current coil B decreases immediately after the end of discharge duration at 1.3 ms, while coil A and coil C keep increasing until 2 ms.

Under the high diluted condition in Figure 13, the initial flame area growth is slower than the basic condition. Furthermore, high current coil B firstly has a faster speed of initial flame growth than that of the standard coil A. After the end of the discharge duration of coil B (1.3 ms), the growth speed decreases. However, the flame area of coil B is reduced to less than coil A until 2 ms under such high diluted conditions. The flame luminance of coil C has the same trend as that of coil B and is less than coil A at 2 ms. Therefore, coil B, with the shorter discharge duration, gets the worst ignition performance under such conditions.

As the image results are shown under the high flow condition, the initial flame has out of the view scope after 1 ms, so that we investigate the flame area and flame luminance before 1 ms under such conditions. The results show the initial flame area keeps increasing rapidly until 1 ms for all the coils in Figure 14, although standard coil A with a lower discharge current has the smallest flame area. Furthermore, the mean luminance of the initial flame for standard coil A starts to decrease after 0.5 ms, which can infer that the enhanced flow increased the speed of the initial flame growth, although this also increased heat transfer between the initial flame kernel and unburned gas. The initial flame kernel generated by the low discharge current will disperse or even be eliminated. Thus, the flame

luminance starts to decrease after 0.5 ms, so that coil A, with the lower discharge current, gets the worst ignition performance under such conditions.

### 4. Conclusions

This study investigated three types of ignition coils under different experimental conditions in a small CVCC, two types of ignition coils with the same discharge energy were analyzed in particular, and several interest points have been found in the combustion results. As such, a new visualization method using a high-speed infrared camera (FLIR X6900sc) and a conventional high-speed camera (FASTCAM SA-X) simultaneously was pioneered in this research. This method can remove the interference of the preheat zone and detect the initial flame kernel more accurately. The combustion results can be explained by analyzing the images of simultaneous visualization. The key features identified in this study are summarized as below:

1. It is confirmed that the high-speed infrared images can ignore the interference of the preheat zone caused by the discharge channel as the infrared images had hardly any brightness around the spark plug under the no fuel condition, while the discharge channel still formed during the ignition process;

2. Increasing the mixture dilution ratio will decrease the speed of initial flame growth, which results in the decreasing of combustion speed and even increasing the ignition energy cannot reach the combustion speed of low diluted condition. While increasing the flow velocity may improve the combustion speed only if increasing the ignition energy correspondingly;

3. Under the high diluted and low flow conditions, prolonging the discharge duration has more benefits to improve the ignition performance compared with the high discharge current at the same ignition energy. This is a result of the long discharge duration extending the heating time to the gas mixture, which can keep the initial flame growth speed and form the self-sustained flame by holding the initial flame around the spark plug;

4. When comparing the same ignition energy coils under the enhanced flow condition, the discharge current has a significant impact on the ignition performance compared to the discharge duration. The enhanced flow stretches and wrinkles the initial flame kernel, and may prevent the initial flame from forming the self-sustain flame. High discharge current ensures the initial flame generation and propagation.

**Author Contributions:** Funding acquisition, T.K., Y.M. and K.O.; Methodology, K.O.; Project administration, Y.M.; Supervision, T.K. and Y.M.; Writing—original draft, Q.C.; Writing—review and editing, T.K. All authors have read and agreed to the published version of the manuscript.

**Funding:** This research was funded by JSPS KAKENHI grant number JP19K04233 And the APC was funded by JSPS KAKENHI.

**Institutional Review Board Statement:** Not applicable.

**Informed Consent Statement:** Not applicable.

**Data Availability Statement:** The study did not report any data.

**Acknowledgments:** This work was supported by JSPS KAKENHI Grant Number JP19K04233.

**Conflicts of Interest:** The authors declare no conflict of interest.

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
