# Peer review of "Study of Discharge Characteristics on Ignition Performance via High-Speed Imaging in a CVCC"

_applsci, doi:10.3390/app12073280_

Round 1

Reviewer 1 Report

Please see the attached PDF of comments for the authors to consider.

Reviewer 2 Report

The authors conducted a series of interesting experiments to explore the discharge characteristics on ignition performance. The structure of the paper is reasonable and the results is helpful to understand the process of ignition in the chamber. I think this paper could be published after revision. Here is the detailed comments.

  1. The literature review section should be strengthened, especially related studies using PLIF to observe combustion regions.
  2. The study about the ignition process should be further discussed, and the influencing factors should be analyzed surfficently.  
  3. The layout of the figures should be re-arranged, cause they are not clear enough.
  4. That's all.

Reviewer 3 Report

The paper is well presented and the topic very interesting. However, for being accepted, the following shortcomings should be addressed.

Title little long. If possible reduce number of words.

Explain better the originality of the work conducted. In reviewer's opinion, the work is valuable, and more on the effect about the discharge in the case of knocking phenomenon, prechamber application should be commented. Here, "Engine Knock Detection Methods for Spark Ignition and Prechamber Combustion Systems in a High-Performance Gasoline Direct Injection Engine," SAE Int. J. Engines 15(6):2022, a work about knocking. How could be your analysis impact on the analysis conducted?

Some figures are a little hard to read. Improve the quality of them.

Are all figures needed. Delete the one not really effective for the work.

Thanks
